# The Effect of The Daily Mile on Primary School Children’s Aerobic Fitness Levels After 12 Weeks: A Controlled Trial

**DOI:** 10.3390/ijerph17072198

**Published:** 2020-03-25

**Authors:** Maxine de Jonge, Jorien J. Slot-Heijs, Richard G. Prins, Amika S. Singh

**Affiliations:** Mulier Institute; Herculesplein 269, 3584 AA Utrecht, The Netherlands; m.dejonge@mulierinstituut.nl (M.d.J.); j.slot@mulierinstituut.nl (J.J.S.-H.); r.prins@mulierinstituut.nl (R.G.P.)

**Keywords:** physical activity, intervention, school-based, support, students, implementation

## Abstract

The Daily Mile (TDM) is a school-based physical activity intervention encompassing a 15-minute run at least three times per week. This study aimed to determine (1) the effects of performing TDM for 12 weeks on Dutch primary school children’s aerobic fitness levels and (2) if additional personal support for teachers impacted the effectiveness of TDM. Nine Dutch primary schools (*n* = 659 children, grades 5–8) were allocated to a control (no TDM), intervention (12 weeks TDM) or intervention-plus (12 weeks TDM, additional personal support) group. The Shuttle Run Test (SRT) was used to assess aerobic fitness at baseline and follow-up. Data were analyzed using a multiple-imputed dataset and multilevel linear regression models to account for the clustering of students within classes and classes within schools. The regression analyses were adjusted for sex and age. Compared with the control group, significant intervention effects of TDM on SRT score were observed for the intervention group (β = 1.1; 95% CI: 0.8; 1.5) and the intervention-plus group (β = 0.6; 95% CI 0.3; 0.9). Additional personal support had no impact on the effectiveness of TDM. These results suggest that performing TDM at least three times per week for approximately 12 weeks increases primary school children’s aerobic fitness. Additional personal support did not improve the effectiveness of TDM on aerobic fitness within this period. These results contribute to the body of evidence surrounding TDM, but further research is needed regarding long-term implementation of TDM.

## 1. Introduction

Aerobic fitness is an important indicator of children’s health [1,2]. There is a strong body of evidence demonstrating that all patterns of physical activity benefit aerobic fitness [3]. Moderate to vigorous physical activity (MVPA) is more consistently positively related to aerobic fitness than light physical activity. On average, only 30% of children age 5–17 years, across 47 countries worldwide, meet the guidelines of 60 minutes of MVPA per day [4]. Parent-reported data show that 56% of children age 4–11 years in the Netherlands meet these guidelines [5]. However, parents may overestimate, as accelerometer data show that just 2% of Dutch girls and 16% of Dutch boys, aged 10–12 years old, meet the MVPA guidelines [6]. Effective interventions are therefore warranted to increase the number of children who are sufficiently physically active and fit.

Schools can function as an ideal venue for physical activity programs [7] since school attendance is mandatory and children spend a substantial amount of time at school. Burns et al. conducted a meta-regression on a wide variety of school-based physical activity interventions and aerobic fitness, and found that school-based physical activity interventions have significant positive effects on aerobic fitness in primary-school-aged children [8]. Successful implementation is essential to the effectiveness of school-based physical activity interventions and is partially dependent on the willingness of the school personnel to allocate time in the curriculum for the intervention [9]. According to teachers and principals, physical activity interventions should be short in duration, executed by teachers themselves, require no preparation, and be able to be executed spontaneously [10]. Furthermore, previous research has shown that interventions do not need to be complex or long in duration to be effective in improving children’s aerobic fitness levels. Burns et al. concluded from their meta-regression that both the intervention length and the number of intervention components did not modify the effect of the interventions on aerobic fitness [8].

One school-based physical activity intervention that meets the aforementioned teacher criteria is The Daily Mile, an intervention created by a Scottish teacher in 2012 to improve her students’ fitness levels [11]. The Daily Mile encompasses a 15-minute run for the whole class on or around school grounds. The 15-minute run is supposed to take place at least three days per week, during school hours, and in addition to recess and physical education time. According to The Daily Mile’s website, more than 400 schools in the Netherlands are registered participants (February 2020) [11].

Two studies have previously investigated the effects of The Daily Mile on aerobic fitness. Chesham et al. [12] found that performing The Daily Mile for five months increased 20-meter Shuttle Run Test (SRT) scores in 4- to 12-year-old Scottish children. Brustio et al. [13] showed that the 6-minute run test scores of 6- to 9-year-old Italian children significantly improved after three months of performing The Daily Mile. Although data on the effectiveness of The Daily Mile in improving children’s aerobic fitness is promising, the evidence remains limited. Daly-Smith et al. [14] expressed their concerns regarding the small sample and unequal dose–response conditions in the study by Chesham et al. [12]. Brustio et al. [13] investigated children from five schools in one region of Italy, with identical educational curricula and similar facilities. Research on large samples of children in varying contexts is needed in order to draw more definitive conclusions about the effectiveness and generalizability of The Daily Mile in improving children’s aerobic fitness.

Other research on The Daily Mile has focused on the implementation and perceived effects of The Daily Mile from the teacher’s perspective. Teachers indicate that the simplicity, autonomy, flexibility, and adaptability in intervention delivery are relevant factors for successfully implementing The Daily Mile [15]. Teachers perceive The Daily Mile as beneficial for their students’ health and fitness [16]. In practice, there is a large variation in implementation between schools [17,18]. Not all schools adhere to the minimum performance frequency (three times per week), duration (15 minutes), and intensity (moderate to vigorous intensity) when performing The Daily Mile. Adherence to The Daily Mile protocol is important for the effectiveness of the intervention in improving children’s aerobic fitness. The main reasons why schools stop performing The Daily Mile are that they have other priorities and that they experience a lack of support [18]. Brustio et al. [13] reported that only half of the teachers (56%) found it easy to perform The Daily Mile without external support. Therefore, adding external support to The Daily Mile may benefit proper implementation and prevent dropout. However, the impact of external support on the effects and implementation of The Daily Mile has not been studied before. 

This study aimed to determine (1) the effects of performing The Daily Mile for 12 weeks on the aerobic fitness levels of Dutch primary school children and (2) if additional personal support for teachers impacted the effectiveness of The Daily Mile.

## 2. Materials and Methods

### 2.1. Study Design

This study was a multi-arm, partly randomized controlled trial with three groups: a control group (regular curriculum), an intervention group (The Daily Mile on the days without physical education class) and an intervention-plus group (The Daily Mile on the days without physical education class, plus additional personal support for teachers). We instructed schools assigned to the control group to follow their regular school curriculum. We provided verbal and written instruction to the schools randomized to the intervention group and the intervention-plus group to perform The Daily Mile on all days without regularly scheduled physical education class (three or four days per week). Schools in the control group did not perform The Daily Mile during the study period but were given the option to start The Daily Mile after the study was completed. We conducted the baseline measurements in March and April of 2019. Follow-up measurements were 12 weeks later, in June and July of 2019. 

The Medical Research Involving Human Subjects Act (WMO) in the Netherlands does not apply to the present study [19], since the present study does not require participants to perform any behaviors that infringe upon their physical or mental integrity. Therefore, no approval from an official ethics committee was required. In addition to the informational material (see recruitment, Section 2.3), we distributed informed consent forms for study participation. We informed parents/caregivers and children that they were able to withdraw from the study at any moment without indicating a reason for their withdrawal. Only children with written permission of their parent/caregiver were allowed to participate in the measurements. 

### 2.2. Intervention Design

Schools in the intervention group participated in the ‘standard’ version of The Daily Mile program (Table 1), in which schools received a welcome packet and performed The Daily Mile three or four times per week. Schools in the intervention-plus group participated in the ‘expanded’ version of The Daily Mile program (Table 1), in which schools received a welcome packet, performed The Daily Mile three or four times per week, and received additional personal support. 

JOGG (Youth at A Healthy Weight), the organization that coordinates and promotes The Daily Mile, provides all schools in the Netherlands that register for The Daily Mile with a welcome packet. The welcome packet contains a poster to show the school is participating in The Daily Mile, temporary tattoos with The Daily Mile logo for the children, informational flyers for parents, an instruction manual for the school’s The Daily Mile coordinator, instruction manuals for teachers, and a calendar that teachers use to mark the days the class performs The Daily Mile. 

The Daily Mile is originally intended to be performed at a minimum frequency of three days per week, in addition to recess and physical education class. We further specified these instructions for schools participating in the present study by asking schools to perform The Daily Mile on all days with no physical education class. If performed according to protocol, students would be physically active during every school day. Depending on their physical education schedule, schools in the present study performed The Daily Mile three or four times per week.

A JOGG employee provided teachers in the intervention-plus group with additional personal support. Support consisted of a personal visit to all schools in the intervention-plus group within the first two weeks after schools started the intervention. The JOGG employee contacted teachers in the intervention-plus group two to three times per week via WhatsApp to remind the teachers to implement The Daily Mile and to ask for updates regarding the implementation. These messages were motivational and solution-oriented. For example, the JOGG employee sent the weather forecast for the week, along with suggestions for when to perform The Daily Mile. At the end of each week, the JOGG employee asked the teachers how many days they had performed The Daily Mile that week. Additionally, the JOGG employee called the teachers every three weeks to discuss possible barriers and other issues related to implementation of the intervention more in-depth, such as weather, motivation, and difficulties transitioning between classroom time and The Daily Mile. If the teachers did not meet the required implementation guidelines (i.e., performing The Daily Mile on the days with no physical education), the JOGG employee called every two weeks.

### 2.3. Recruitment

A priori power calculations showed that 175 participants per group (525 in total) were needed to detect a 0.5 step difference in the Shuttle Run Test (SRT) outcomes between groups (alpha: 0.05; beta: 0.80).

We recruited schools between February 2019 and mid-April 2019 through (1) JOGG and (2) our professional and personal networks. Approximately 100 schools across the Netherlands were contacted for participation, 14 of which were interested in study participation (Figure 1). 

We contacted schools that showed interest in participating in the current study via email and telephone and provided them with a standard informational letter about The Daily Mile, as well as the aim and methods of the study. 

During the recruitment phase, we asked schools if they were willing to perform The Daily Mile as a part of their daily curriculum for 12 consecutive weeks in the period between March and July 2019, excluding the spring vacation period. Spring vacation was either one or two weeks long, depending on the school’s schedule. Schools that had any prior experience implementing The Daily Mile were ineligible for participation in the study. Schools were also ineligible if it was not possible to schedule a follow-up measurement 12 school weeks from the baseline measurement, and if they had less than two classes willing to participate in the study. Schools unwilling to perform The Daily Mile for 12 weeks in that time period were allocated to the control group. We randomly allocated the remaining schools to the intervention group or intervention-plus group. Researchers and research assistants present at the baseline and follow-up measurements were not blinded to the school’s allocation. We notified schools about their intervention group assignment after their inclusion in the study but prior to their baseline measurement. 

### 2.4. Outcomes

The baseline measurement for all groups consisted of the SRT. Parents provided their child’s date of birth and sex on the same form as the informed consent. Age was calculated as the age in years on the day of the baseline measurement. 

To assess aerobic fitness, we administered the SRT according to the EuroFit protocol during a regularly scheduled gym class [20]. The SRT has been shown to be a reliable, valid, and feasible field test for estimating maximal oxygen uptake (VO_2_max) in 8- to 19-year old children in the school setting [21,22]. The SRT consists of a number of stages, each lasting one minute, paced by beeps at preset intervals. As the test proceeds, the interval between each successive beep reduces, forcing participants to increase their running speed. A standard SRT recording with music was used for all measurements. Due to the limited size of some of the school gymnasiums, all SRTs were conducted on a standard volleyball court (18 meters in length instead of 20 meters, which is standard protocol). Participants ran the 18-meter course back and forth, in groups of ten to twelve participants, and started at a 7.2 km/hour pace. Two administrators (one researcher, one research assistant) were present at any given measurement. In total, nine people (two researchers, seven research assistants) administered SRTs throughout the study. We provided participants with standardized instructions. The SRT was over when a participant stopped the test voluntarily, or when they were more than two meters away from the end line when the beep sounded, two consecutive times. A participant’s score was the level last announced on the recording. Researchers did not encourage the participants during the test, but fellow children and teachers were permitted to do so. 

Baseline measurements were conducted in March and April of 2019, with follow-up measurements in June and July of 2019. 

The follow-up measurement consisted of a second SRT. The protocol for the SRT was identical to that at baseline. At least one of the two administrators present at the baseline test was also present at the follow-up test. Children were tested in the same groups as at baseline.

Intervention fidelity was measured with self-report data. Teachers marked the days that the class performed The Daily Mile on the calendar included in their welcome packet. 

### 2.5. Analyses

Since the primary aim of the study was to determine the effect of The Daily Mile on aerobic fitness, we pooled the intervention group and intervention-plus group data into a ‘combined intervention’ group.

In line with the first aim of the study, we performed descriptive statistics on all outcome variables (baseline SRT score, follow-up SRT score, and change in SRT score) for the control group and the intervention groups combined. When outcome variable significantly differed between the control group and the combined intervention group, we conducted post-hoc testing on the intervention and intervention-plus group. 

Intervention duration was calculated as the number of eligible intervention weeks between baseline and follow-up. This did not include school vacation week(s). 

Implementation rate was calculated as the number of days the class performed The Daily Mile divided by the total number eligible for The Daily Mile performance, multiplied by 100. 

The use of multiple-imputation techniques to impute missing outcome and covariate values is a recommended strategy to minimize biased effect estimates [23]. We used multiple imputation to impute missing covariates and outcomes at baseline and follow-up. These values were imputed in 20 datasets with chained equations, using multivariate normal regressions to impute missing values. See Table A1 in Appendix A for the variables and number of participants per variable for whom data were imputed. 

Our data had a multilevel structure in which students were clustered within classes and classes were clustered within schools. Therefore, we applied a multilevel linear regression model to assess the effects of The Daily Mile on aerobic fitness (SRT score). In this model, observations were clustered within individuals, and individuals were clustered in classes and schools. Time, group, and the time*group interaction were regressed on the outcome (SRT score) in this model. The time*group interaction term denotes the differences observed between groups over time. All models were additionally adjusted for sex and age. Stata 13.1 was used for statistical analyses (StataCorp LP, College Station, TX, USA).

## 3. Results

### 3.1. Participation and Study Completion Rate

Nine schools participated in the study, and 659 eligible children provided informed consent (Figure 1). The median number of classes per school was four (range 2–7 classes). A complete data set was obtained for 536 participants, i.e., age, gender, and SRT score at baseline and follow-up. Absence was the primary reason for missing data. 

### 3.2. Participant Baseline Characteristics

Baseline characteristics of the study participants with complete, non-imputed baseline data (age, sex, and SRT score) are presented in Table 2. The mean age of the intervention group (mean 9.7 years; Standard Error (SE): 0.1)) was significantly lower than the control group (10.1; SE: 0.1; *p* < 0.01) and intervention-plus group (10.1; SE: 0.1; *p* < 0.05). The mean baseline SRT score was significantly higher in the control group (mean 7.6 stages; 95% Confidence Interval (CI): 7.4; 7.9) when compared with the intervention-plus group (5.9; 95% CI: 5.6; 6.2), the intervention group (5.8; 95% CI: 5.3; 6.2), and the combined intervention group (5.9; 95% CI: 5.6; 6.1).

### 3.3. Change in SRT Score

SRT scores for all time points are shown in Table 3. After adjusting for age and sex, the adjusted model showed a significant intervention effect on SRT score after 12 weeks for both the intervention group (1.1 stages; 95% Confidence Interval (CI): 0.8; 1.5) and the intervention-plus group (0.6; 95% CI: 0.3;0.9), when compared with the control group. The change in SRT score observed in the intervention group was significantly greater than the change observed in the intervention-plus group.

### 3.4. Intervention Fidelity

Schools were asked to perform The Daily Mile on the days with no physical education class. Depending on their physical education schedule, schools in the present study were expected to perform The Daily Mile three or four times per week. All except one of the schools in the intervention group had physical education classes scheduled twice per week and had to perform The Daily Mile three times per week. The other school in the intervention group had physical education class once per week and had to perform The Daily Mile four times per week. Classes in the intervention group had a mean implementation rate of 88%. All schools in the intervention-plus group had physical education classes scheduled twice per week and therefore only needed to perform The Daily Mile three times per week. Classes in the intervention-plus group had a mean implementation rate of 90%. The difference in implementation rates between the intervention and intervention-plus group was not significant. 

### 3.5. Intervention Duration

The mean time between the baseline and follow-up measurements for all groups was 11.6 weeks (standard error (SE): 0.0). The mean time between baseline and follow-up measurements was 11.1 (SE: 0.0) weeks for schools in the control group, 12 (SE: 0.0) weeks for classes in the intervention group, 12.2 (SE: 0.0) weeks for schools in the intervention-plus group, and 12.2 weeks (SE: 0.0) for schools in the intervention groups, combined. Differences in average time between baseline and follow-up measurements between groups were not significant.

## 4. Discussion

In the current study, we assessed the effects of performing The Daily Mile for 12 weeks on children’s aerobic fitness, and if additional personal support for teachers impacted the effectiveness of The Daily Mile. Our results show that performing The Daily Mile at least three times per week for approximately 12 weeks increases 8- to 12-year-old children’s aerobic fitness. Additional personal support did not improve the effectiveness of The Daily Mile on aerobic fitness within this period. The aerobic fitness of the group without additional personal support (intervention group) increased significantly more than the aerobic fitness of the group with additional personal support (intervention-plus group). Both the intervention group and the intervention-plus group reported relatively high implementation rates. 

Our results are in line with those of other recent controlled trials examining The Daily Mile’s impact on children’s aerobic fitness [12,13]. Brustio et al. [13] used the same intervention period (i.e., 12 weeks) as our study and also included a large sample of children. However, methodological differences prevent us from directly comparing our data with that of Brustio et al. [13], since they assessed aerobic fitness using the 6-minute run test. Moreover, their participants were slightly younger (6–9 years old) than the participants in our study (8–12 years old). Differences in study design (i.e., small sample sizes and unequal dose–response time between the control group and intervention group) also prevent us from directly comparing our results with those of Chesham et al. [12]. Contrary to our study, neither Chesham et al. [12] nor Brustio et al. [13] provided schools with implementation instructions regarding a minimal performance frequency for The Daily Mile. Brustio et al. [13] reported that schools performed The Daily Mile on average three times a week, and Chesham et al. [12] did not report the performance frequency. Therefore, the extent to which the implementation of The Daily Mile differs across the three studies is unclear. Regardless of the different methodologies and contexts, the results of all three studies suggest that The Daily Mile can yield beneficial effects on children’s aerobic fitness levels within a short time period. 

One of the novel components of our study, with respect to the other effect studies on The Daily Mile, was the additional personal support for teachers in the intervention-plus group, to aid in the implementation of The Daily Mile. Coaching and support are regarded as core components of intervention implementation [24,25] and are related to intervention effectiveness [24,26]. Reinke et al. found that the amount of coaching teachers received over time was positively correlated with the implementation of a classroom management intervention [27]. Our results show that the additional personal support for teachers did not benefit intervention effectiveness or implementation over a 12-week period. Both the intervention and intervention-plus group had a relatively high intervention implementation rate. However, previous research has shown that adherence to the intended protocol for The Daily Mile can vary considerably [17,18]. This suggests that additional personal support may be superfluous for intervention implementation in the first 12 weeks of the intervention period. It is also possible that schools in the intervention group without additional personal support were subject to the Hawthorne effect: if they had participated in The Daily Mile outside of a research context, they might have adhered less strictly to the protocol than the schools with additional personal support [28]. It is also possible that the current study design was inadequately suited to assessing the impact of additional personal support and that the effects of additional personal support may only manifest over a longer period of time (i.e., longer than 12 weeks). 

A main strength of our study was the large sample of children (*n* = 659) from nine different schools spread across the Netherlands. Schools from urban, suburban, and rural areas were represented in the current study, increasing the generalizability of our findings. Furthermore, we standardized several study aspects: (1) we performed baseline and follow-up measurements according to standardized protocols; (2) in order to minimize the influence of external factors, at least one of the two researchers present at the baseline measurement was also present at the follow-up measurement; (3) in order to minimize the impact of seasonal and weather factors, we conducted all baseline and follow-up measurements in the same time period and within a relatively short period of time [29,30,31]; (4) the additional personal support for the intervention-plus group was provided according to a standardized protocol by a single person, and was therefore the same for all schools in the intervention-plus group, minimizing bias and ensuring equal treatment. 

The study was limited by the partly randomized design. Schools elected to be in the control or one of the intervention groups, which may have created some level of bias. However, the allocation to the intervention and intervention-plus group was randomized. Although we adjusted our analyses for age and gender, there may have been unmeasured confounders, such as BMI or physical activity outside of school, that may have affected our outcomes. 

## 5. Conclusions

This study demonstrates that The Daily Mile, a relatively simple, school-based physical activity intervention, can yield a beneficial effect on children’s aerobic fitness. With an additional 15 minutes of physical activity at least three days per week, primary school children can significantly increase their aerobic fitness levels within a 12-week period. These results contribute to the body of evidence for The Daily Mile as an effective tool for improving children’s aerobic fitness. 

Our study provides several directions for future studies. First, the dose–response relationship between aerobic fitness and the intensity and duration of The Daily Mile performance should be objectively investigated, to better understand optimal intervention implementation. In addition, it is important to establish if there are subgroups of children for which The Daily Mile is more effective, for example, overweight children or children with lower activity levels. Furthermore, in light of the hypothesized effects of The Daily Mile on children’s mental, emotional, and social health, future studies should assess the effects of The Daily Mile on these health outcomes. Finally, it is essential to conduct studies on The Daily Mile with a longer intervention period, for example, an entire school year, in order to determine the long-term effects on aerobic fitness and to uncover what is needed for sustainable intervention implementation. Both student’s and teacher’s perspectives should be taken into consideration in studies concerning sustainable intervention implementation.

## Figures and Tables

**Figure 1 ijerph-17-02198-f001:**
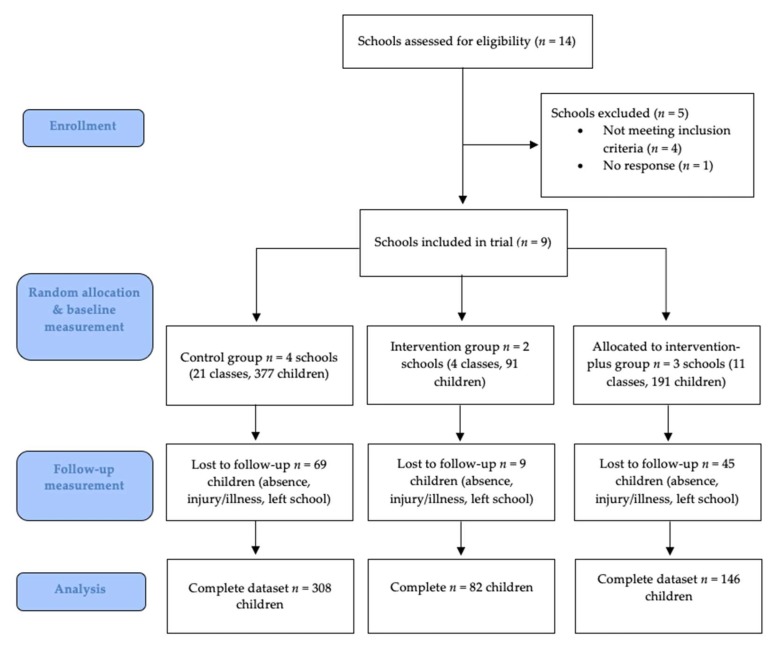
Flow diagram of participants and study design (CONSORT 2010 Flow Diagram).

**Table 1 ijerph-17-02198-t001:** Intervention components by experimental condition.

	Control Group	Intervention Group	Intervention-Plus Group
Welcome packet		✓	✓
Perform The Daily Mile 3 or 4 times per week		✓	✓
Additional personal support			✓

**Table 2 ijerph-17-02198-t002:** Baseline characteristics of study participants for all covariates and outcomes by experimental condition.

	Control Group (*n* = 338)	Combined Intervention Group (*n* = 253)	Intervention Group (*n* = 86)	Intervention-Plus Group (*n* = 167)
Sex (% female)	52.1	50.2	48.8	50.9
Age at baseline (mean years (SE))	10.1 (0.1)	10.0 (0.1)	9.7 (0.1) *^,^^◊^	10.1 (0.1)
Shuttle Run Test (mean stages (95% CI))	7.6 (7.4; 7.9)	5.9 (5.6; 6.1) *	5.8 (5.3; 6.2) *	5.9 (5.6; 6.2) *

SE = Standard Error, CI = confidence interval, * significant difference from control, *p* < 0.01, ◊ significant difference from intervention-plus group *p* < 0.05.

**Table 3 ijerph-17-02198-t003:** Baseline, follow-up, and difference in change in between groups in Shuttle Run Test scores ^¥^, by experimental condition.

	Baseline, Mean (95% CI)	Follow-Up, Mean (95% CI)	Difference in Change between Groups, Beta ^a^ (95% CI)
Control group (*n* = 377)	7.6 (7.4; 7.8)	7.7 (7.5; 7.9)	REF
Intervention groups, combined (*n* = 282)	5.9 (5.6; 6.1)	6.7 (6.5; 7.0)	0.8 (0.5; 1.0) *
Intervention group (*n* = 91)	5.7 (5.6; 6.2)	6.9 (6.4; 7.4)	1.1 (0.8; 1.5) *
Intervention-plus group (*n* = 191)	5.9 (5.6; 6.2)	6.6 (6.3; 7.0)	0.6 (0.3; 0.9) *^,^^◊^

^¥^ Multiple-imputed values, CI = confidence interval, ^a^ regression coefficient (adjusted model), * significant difference from control, *p* < 0.01, ^◊^ significant difference from intervention-plus group *p* < 0.05.

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
