# Peer review of "The Effect of The Daily Mile on Primary School Children’s Aerobic Fitness Levels After 12 Weeks: A Controlled Trial"

_ijerph, 2020, doi:10.3390/ijerph17072198_

Round 1
Reviewer 1 Report
Thank you for providing the opportunity to review this manuscript. The authors analyzed the effect of the Daily Mile on primary school children's aerobic fitness level after 12 weeks. The manuscript is interesting, well written and presented and overall it is a smooth read. I do not have any major concerns but some minor concerns/suggestions are below:
- The introduction should provide an overview of previous studies not only on The Daily Mile and Netherland but other approaches and in other areas.
- How to detect the children's activity outside of the schools? Some children might be enrolled in off-school activities that might have impacted their aerobic fitness? Perhaps, it should be mentioned as a limitation.
- While asking for consent, could it be possible to fill out a small survey by parents or guardians on child's BMI and about child s enrollment in any other physical activity outside of the school?
Reviewer 2 Report
The problem of aerobic fitness in children and youth is important, and therefore any information about possible benefits of certain programs on improvement of this fitness component is welcomed. In this study authors provided one specific insight into effectiveness of the 1-mile-run program with additional "intervention" on changes in aerobic fitness of the Dutch children.- Although paper is generally well done I must say that statistical approach is currently difficult to follow, and (as far as my opinion is concerned) unnecessary complicated. Please see later for details
Abstract: Several sentences start with "We, etc." Please rewrite it.
Keywords: Avoid terms used in Title
Introduction: This is the best part of the manuscript. Clear, concise and logically resulting in study aim. My congratulations.
Methods: My main concern is related to statistical analysis and presentation of the result. I would strongly suggest authors to use standard 2-way Anova (time x group) and to report results of such hypothesis based statistics in explaining the (eventual) differential effects of the applied programs. If they are concerned about (eventual) initial differences in aerobic fitness (i.e. it seems that control group is superior when compared to other two groups in their initial (pre-testing) capacity) then please consider ancova calculation. I'm not trying to argue that statistical approach is not correct (maybe it is, although I can not be certain), but rather try to point that convenient procedure will be more applicable and undestandable.
Discussion needs additional work. Why did you start your discussion with "limitations and strengths"? Better put it at the end of Discussion section. I will strongly suggest you to start the discussion with main findings, and then to discuss those main findings thoroughly. Otherwise, text is scattered and not connected. You are jumping from one point to another, with no evident connection.
The conclusion needs systematic rewriting. I'd rather see directions for future studies in this part of the text. Also, in the Conclusiob you should provide Conclusions and not results (as it is now).
Reviewer 3 Report
Please change "compared to" to "compared with" throughout the manuscript.
The entire manuscript is well written and clear.
While this research is carefully designed and the manuscript is very clearly written, I think the findings are of low interest to the research community. It is interesting that the added support for staff had no effect. I suggest that this is because it is a school based programme and adherence would was very high. I think running the programme for children who do not remain in sports programmes would be a more interesting target demographic for future research.
Round 2
Reviewer 2 Report
Thank you for your explanations and amendments